# Revealing Nanodomain Structures of Bottom-Up-Fabricated Graphene-Embedded Silicon Oxycarbide Ceramics

**DOI:** 10.3390/polym14173675

**Published:** 2022-09-04

**Authors:** Dongxiao Hu, Gaofeng Shao, Jun Wang, Aleksander Gurlo, Maged F. Bekheet

**Affiliations:** 1School of Chemistry and Materials Science, Institute of Advanced Materials and Flexible Electronics (IAMFE), Nanjing University of Information Science and Technology, Nanjing 210044, China; 2Chair of Advanced Ceramic Materials, Institute of Materials Science and Technology, Faculty III Process Sciences, Technische Universität Berlin, Straße des 17. Juni 135, 10623 Berlin, Germany

**Keywords:** graphene, ceramic, graphene aerogel, silicon oxycarbide, nanodomain

## Abstract

Dispersing graphene nanosheets in polymer-derived ceramics (PDCs) has become a promising route to produce exceptional mechanical and functional properties. To reveal the complex nanodomain structures of graphene–PDC composites, a novel reduced graphene oxide aerogel embedded silicon oxycarbide (RGOA-SiOC) nanocomposite was fabricated bottom-up using a 3D reduced graphene oxide aerogel as a skeleton followed by infiltration of a ceramic precursor and high-temperature pyrolysis. The reduced graphene oxide played a critical role in not only the form of the free carbon phase but also the distribution of SiO_x_C_4−x_ structural units in SiOC. Long-ordered and continuous graphene layers were then embedded into the amorphous SiOC phase. The oxygen-rich SiO_x_C_4−x_ units were more prone to forming than carbon-rich SiO_x_C_4−x_ units in SiOC after the introduction of reduced graphene oxide, which we attributed to the bonding of Si atoms in SiOC with O atoms in reduced graphene oxide during the pyrolysis process.

## 1. Introduction

Graphene-polymer-derived ceramics (PDCs) nanocomposites are an emerging material system actively researched in advanced structural and functional fields. Proper integration between graphene and PDC is of high practical value, taking advantage of the beneficial properties of both. For instance, adding PDCs to graphene can improve its thermal stability, especially at high temperatures [1,2]. PDCs can be mechanically strengthened by graphene fillers and endowed with multiple functions, such as thermal, electrical, or shielding properties. A series of PDCs (e.g., SiOC [3,4,5,6,7,8,9,10], SiCN [2,11,12,13], SiBCN [1,14], SiOCN [15,16], and Si_3_N_4_ [17]) matrix composites containing graphene filler have been obtained so far. For example, graphene nanosheets serve as a reinforcement phase for improving the mechanical properties of SiOC [4,6] or SiC [18]. Graphene–SiOC nanocomposites are being explored as stable anodes for lithium-ion batteries, using the synergistic contribution of graphene for rapid electron transfer and SiOC for robust electrochemical Li^+^ ion storage [3,7,9,19]. The graphene–SiOC, –SiCN, –SiBCN, or –Si_3_N_4_ nanocomposites with good oxidation resistance have been recently applied as high-temperature electromagnetic wave-absorbing material under harsh environments [1,10,11,17]. Additionally, a graphene–SiOCN composite was used for thermal management applications through surface modification of graphene with electrical insulated SiOCN coating [15].

However, there are two main challenges in the field of graphene–PDC composites that have not yet exhibited substantial progress. First, controllable dispersion of the graphene nanosheets, especially with high loadings, in the PDCs is extremely difficult using conventional top-down physical mixing or chemical bonding methods due to the strong van der Waals force between the 2D graphene nanosheets. Thus, graphene tends to agglomerate in the PDC matrix, deteriorating the mechanical, electrical, or shielding properties of the composites. The second challenge is that the nanodomain structures of graphene–PDC material systems, which are critical in clarifying structure–property relationships and ultimately in determining their properties for specific engineering applications, are not yet fundamentally understood [20]. Most work has mainly focused on the increased free carbon phase of graphene–PDC systems, and little attention has been paid to the effect of graphene oxide on the structure of Si-based nanodomains within PDCs. For example, SiOC generally possesses a complex nanostructure in which corner-shared SiO_x_C_4−x_ (0 ≤ x ≤ 4) tetrahedral structures are surrounded by a fraction of free carbon (C_free_) [21,22,23]. This unique network was extensively documented with various spectroscopic, scattering, and electron microscopic techniques [20], while few reports have focused on revealing the nanodomain structure of the graphene–SiOC hybridized nanocomposites. It is difficult to distinguish between the nanodomains of the graphene–PDC composite prepared by the top-down approaches due to the lack of a fine and homogeneous structure.

In this work, we put forward a universal and delicate bottom-up processing strategy to uniformly disperse graphene into SiOC ceramic by using a 3D reduced graphene oxide aerogel as a skeleton to infiltrate polysiloxane precursors. This strategy has great potential to enable both mechanical reinforcement and multifunctionalization, to which a more controllable design of the geometrical morphology and material constituents is fundamentally attributed. The nanodomain structures, especially C_free_ and SiO_x_C_1−x_ tetrahedral units of reduced graphene oxide embedded SiOC nanocomposites, were investigated using several characterization techniques, including HR-TEM, Raman, XPS, and NMR. The insights into the nanodomain structure of graphene–SiOC composites allow for a greater understanding of structure–property relationships and describe an efficient pathway for designing high-performance graphene–PDCs nanocomposites.

## 2. Materials and Methods

### 2.1. Preparation of 3D Porous Reduced Graphene Oxide Aerogel Skeleton

Three-dimensional reduced graphene oxide aerogel (RGOA) was synthesized using a hydrothermal method, followed by freeze-drying and thermal reduction. First, graphene oxide (GO) was prepared according to our previous work [24] and then ultrasonically dispersed in cold deionized water (5 mg GO/mL) for 2 h. An aqueous mixture of pyrrole (Py) monomer and GO suspension with a weight ratio of Py:GO = 5:1 was ultrasonically dispersed for 30 min and then transferred into a Teflon-lined autoclave at 150 °C for 6 h. The resulting product, a black reduced graphene oxide hydrogel (RGOH), was then rinsed with a water/ethanol solution with a volume ratio of 5:1 for 24 h to remove residual Py and polypyrrole (PPy). The reduced graphene oxide polypyrrole (RGO/PPy) aerogel was obtained by freeze-drying the RGOH for 24 h. RGO/PPy aerogel was further thermally reduced at 600 °C and 1000 °C for 1 h in a tube furnace under flowing argon at a heating rate of 3 °C/min to combust the organic content and form reduced graphene oxide aerogel (RGOA). The aerogels obtained after thermolysis at 600 and 1000 °C are denoted hereafter as RGOA-600 and RGOA-1000, respectively.

### 2.2. Preparation of Reduced Graphene Oxide Embedded Silicon Oxycarbide Nanocomposites

Commercial preceramic polymer precursor methylphenylvinylhydrogen polysiloxane (SILRES H62C, Wacker Chemie, Munich, Germany) was infiltrated into the as-prepared RGOA frameworks under vacuum to fabricate 3D graphene-polymer-derived ceramic architectures. We prepared 25 mg/mL polymeric solution by dissolving 100 mg polymeric precursor into 4 mL tert-butanol (TBA, (CH_3_)_3_OH); RGOA samples (RGOA-600 and RGOA-1000) were then immersed into this solution under vacuum for 2 h to facilitate precursor penetration into the aerogel and eliminate bubbles. Subsequently, the polymeric precursor-loaded RGOA (hereafter denoted as RGOA-600-P and RGOA-1000-P) samples were removed from the solution and freeze-dried for 12 h to remove the TBA. The obtained RGOA-P cylinders were further cross-linked at 250 °C for 2 h and pyrolyzed at 1000 °C for 3 h in a tube furnace under flowing argon at a heating rate of 100 °C/h to obtain the final 3D RGOA-SiOC materials. The obtained ceramic nanocomposite materials after pyrolysis of RGOA-600-P and RGOA-1000-P are denoted hereafter as RGOA-600/SiOC and RGOA-1000/SiOC, respectively. The final weight ratio (initial weight of RGOA/ weight of RGOA-SiOC nanocomposite) of RGOA in RGOA-600/SiOC and RGOA-1000/SiOC was 22.3 and 20.5 wt%, respectively. For comparison, unmodified SiOC material was prepared by the pyrolysis of preceramic polymer H62C at the same heat-treatment conditions.

### 2.3. Characterization

The morphology and microstructure of RGOA and RGOA-SiOC composites were investigated with scanning electron microscopy (SEM) using a Leo Gemini 1530 and transmission electron microscopy (TEM) using a JEOL 2100F. X-ray photoelectron spectroscopy (XPS) was performed on a Thermo Fisher Scientific ESCALAB 250Xi to investigate the surface chemical states of different elements in SiOC and reduced graphene oxide modified SiOC nanocomposites. Raman spectra of RGOA and RGOA-SiOC composites were measured using a micro-Raman spectrometer (RENISHAW inVia) with an excitation wavelength of 532 nm. Solid-state -^29^Si DD/MAS NMR spectra were recorded on an Agilent 600 DD2 spectrometer (Agilent, Santa Clara, CA, USA, magnetic field strength 14.1 T) at a resonance frequency of 199.13 MHz for 29 ^29^Si using dipole decoupling magic angle spinning (DD/MAS) and high-power 1H decoupling. The powder samples were placed in a pencil-type zirconia rotor with an outer diameter of 4.0 mm. The spectra were obtained at a spinning speed of 8 kHz (4 μs 90° pulses) and a recycle delay of 10 s. The Si signal of tetramethylsilane (TMS) at 0 ppm was used as the reference ^29^Si chemical shift. The scanning number was 5000. Solid-state ^13^C CP/MAS NMR spectra were recorded on an Agilent 600 DD2 spectrometer (Agilent, Santa Clara, CA, USA, magnetic field strength 14.1 T) at a resonance frequency of 150.72 MHz for ^13^C using cross-polarization (CP), magic-angle spinning (MAS), and high-power 1H decoupling. The powder samples were placed in a pencil-type zirconia rotor with an outer diameter of 4.0 mm. The spectra were obtained at a spinning speed of 10 kHz (4.2 μs 90° pulses), a 2 ms CP pulse, and a recycle delay of 3 s. The C signal of tetramethylsilane (TMS) at 0 ppm was used to reference the [13] C chemical shift.

## 3. Results and Discussion

Three-dimensional reduced graphene oxide aerogels (RGOAs) were prepared from GO and PPy via a typical hydrothermal process at 150 °C for 6 h, followed by freeze-drying for 24 h and thermal reduction at 600–1000 °C for 1 h, as illustrated in Figure 1. During the hydrothermal process, the PPy preferentially grows on the surface of GO sheets due to electrostatic interactions between positively charged PPy and negatively charged GO surfaces and π–π interactions between PPy rings and conjugated segments in GO and hydrogen-bonding interactions, resulting in the reduction of GO and the formation of RGO-PPy aerogel [19,24]. Taking advantage of in situ crosslinking of the PPy, GO nanosheets can be aligned along the flow direction during the hydrothermal process (Figure 2a) [25]. During further thermal reduction of RGO-PPy aerogels at 600 and 1000 °C in Ar, PPy decomposes, yielding RGOA with different degrees of reduction. As shown in Figure 2b, the aerogel retained the aligned macropore structure after thermal reduction, which provides sufficient space for the formation of the preceramic precursor and promotes its infiltration into the RGOA skeleton. RGOA/SiOC nanocomposites were then fabricated by precursor infiltration, freeze-drying, and high-temperature pyrolysis (Figure 1). During infiltration, the H62C precursor attaches to RGOA surfaces due to its favorable wetting and adhesion ability and π–π interactions between RGOA and phenyl groups in the H62C precursor. Tert-butanol (TBA) was chosen as the solvent in the infiltration step instead of other organic solvents (e.g., ethanol or tetrahydrofuran) to overcome the problem of surface tension at the gas–liquid–pore wall during the subsequent freeze-drying process. As a result, the 3D graphene skeleton maintained its morphology without shrinkage, collapse, or agglomeration during infiltration and drying. As shown in Figure 2c,d, the obtained RGOA/SiOC nanocomposite after further pyrolysis at 1000 °C was composed of layered graphene-embedded SiOC ceramic, which was more suitable for the further investigation of nanodomains within graphene-SiOC composite than SiOC-based particles with randomly distributed graphene.

The TEM images of SiOC pyrolyzed at 1000 °C (Figure 3a,b) show that SiOC ceramic is amorphous and homogenous. The size of the free carbon phase in SiOC pyrolyzed at 1000 °C was rather small, and the so-called basic structural units (BSUs) of the free carbon with few lamellar carbon layers could be detected even at higher magnification (Figure 3b). In contrast, after introducing SiOC into RGO aerogel, a long-ordered carbon phase was embedded in the amorphous SiOC phase (Figure 3d,e), a structure that has not been observed in other similar graphene-modified polymer-derived ceramics [12,13,16].

Micro-Raman spectroscopy was used to analyze the structure of the carbon phase within reduced graphene oxide aerogels, SiOC, and reduced graphene oxide embedded SiOC samples. Figure 4a shows the Raman spectra of the as-prepared RGOAs with different reduction temperatures. The D and G bands observed at 1345 and 1585 cm^−1^ could be attributed to disordered sp^2^-hybridized modes in the carbon rings and in-plane bond stretching of sp^2^ carbon, respectively. In addition to the typical D and G bands (at 1320 and 1600 cm^−1^) in the pure SiOC sample, the 2D, D + G, and 2D’ bands were observed in the second-order Raman spectra at around 2610, 2900, and 3195 cm^−1^, respectively, which could be assigned to the overtones and combinations of different Raman vibration modes in PDC materials [23]. Because the intensity ratio of the D and G modes provides valuable information about the structural arrangement of the free carbon phase present in the SiOC network, the Raman spectra were fitted using the Lorentzian curve fitting for the D1, D4, and G bands and Gaussian curve fitting for the D_3_ band [26,27,28]. As shown in Figure 4c–g and Table 1, the D_4_ band was observed as a shoulder of the D_1_ band at ca. 1190 cm^−1^ in the spectra of all SiOC, RGOA-600, RGOA-1000, and RGOA-SiOC nanocomposites, which could be attributed to the disordered graphitic lattice (C−C and C=C stretching vibrations and sp^2^-sp^3^ bonds) in soot and related carbon materials. The D_3_ band at 1500–1510 cm^−1^ could be assigned to the amorphous carbon fractions [20,23].

The intensity ratio (I_D1_/I_G_) of RGOA, which indicates the reduction degree of graphene oxide, decreased from 1.14 to 1.08 with increasing reduction temperature. This result revealed the removal of defects and recovery of conjugated domains in the reduced graphene oxide, especially at higher thermal reduction temperatures. The extracted I_D1_/I_G_ intensity ratio can also be used to determine the lateral crystallite size of free carbon (the length of the carbon domain along the sixfold ring plane, donated as *La*). Until now, two correlations between I_D1_/I_G_ and *La* have been proposed, as shown in Equations (1) and (2) [23].
I_D1_/I_G_ = C(λ)/*La_1_*(1)
I_D1_/I_G_ = C’(λ)*La_2_^2^*(2)
where C(λ) and C’(λ) are wavelength-dependent prefactors, C(λ) = C_0_ + λC_1_ (C_0_ = −12.6 nm and C_1_ = 0.033), C’(λ) ≈ 0.55 nm^−2^. λ is the wavelength of the laser; and I_D1_ and I_G_ are the intensities of the D_1_ and G band, respectively.

Generally, the Tuinstra and Koenig (TK) correlation (Equation (1)) is valid for carbon clusters with *La* values higher than 2 nm, while the Ferrari–Robertson equation (Equation (2)) is valid for *La* values lower than 2 nm [23]. In the case of SiOC, the *La_2_* value is about 1.48 nm, which is theoretically acceptable. However, the TK correlation is more suitable for evaluating the *La* value of graphene–SiOC nanocomposites, which was 4.63 and 4.81 nm for the RGOA-600/SiOC and RGOA-1000/SiOC samples, respectively.

The elemental composition and chemical environment of C and Si on the surface of the samples were examined by XPS to gain insight into not only the reduction progress of graphene oxide [29], but also the evaluation of SiO_x_C_4−x_ units after the introduction of graphene [30]. As shown in Figure 5a, the intensity of the C 1s peak, located at 284.6 eV, continually increased from GO to RGOA-600 and RGOA-1000, while the change in oxygen content (as indicated by the O 1s peak at ca. 532.4 eV) showed the opposite trend. GO exhibited a C/O atomic ratio of 1.9, which increased to ~9.94 and 25.81 after further thermal reduction at 600 and 1000 °C, respectively, confirming the reduction of graphene oxide sheets. As summarized in Table 2, the highest C/O atomic ratio determined for the RGOA-1000 sample indicated that the reduction eliminated most of the oxygen-containing groups. The surface elemental composition calculated from XPS was found to be 18.6, 34.6, and 46.8 at % for Si, O, and C in SiOC, respectively; the RGOA-600/SiOC and RGOA-1000/SiOC nanocomposites showed lower silicon amounts (i.e., 10.7 and 8.6 at %, respectively), lower oxygen (25.0 and 20.6 at %, respectively), and higher carbon contents (62.5 and 69.0 at %, respectively), with small amounts of nitrogen (1.8 and 1.8 at %, respectively). These results further confirmed that SiOC was successfully integrated into the reduced graphene oxide aerogel.

Figure 5b shows the high-resolution C 1s spectrum of GO, which could be deconvoluted into four peaks assigned to C−C (284.6 eV), C−O (286.6 eV), C=O (287.8 eV), and O-C=O (289.1 eV) [29] bonds and groups. The relative intensity of C−O, C=O, and O-C=O peaks in RGOA decreased, and two peaks corresponding to C−N bond (285.6 eV) and π−π stacking (290.8 eV) appeared after the thermal removal of oxygenated functional groups. The Si 2p peaks in SiOC and RGOA-SiOC nanocomposites (Figure 5c) could be deconvoluted into five peaks assigned to SiO_4_ (103.6 eV), SiO_3_C (102.8 eV), SiO_2_C_2_ (101.8 eV), SiOC_3_ (100.8 eV), and SiC_4_ (99.5eV) groups [30]. It is worth noting that the relative amount of SiC_4_, SiOC_3_, and SiO_2_C_2_ units decreased and the content of oxygen-rich SiO_x_C_4−x_ units (SiO_3_C and SiO_4_) increased after the incorporation of RGOA. Especially in the case of lower reduction degrees of RGOA-600, SiC_4_ and SiOC_3_ units almost disappeared, and the contribution of SiO_2_C_2_ decreased, whereas the proportion of SiO_3_C and SiO_4_ increased. These results suggested that RGOA can tailor the structure of SiO_x_C_4-x_ units in SiOC ceramic by converting the carbon-rich SiO_x_C_4-x_ units into oxygen-rich SiO_x_C_4−x_ units (Figure 5c).

^29^Si MAS-NMR spectroscopy was further used to probe the structure of SiO_x_C_4−x_ tetrahedra present in the amorphous network of SiOC and RGOA-SiOC nanocomposites. Four different environments were evident in the SiOC sample obtained by pyrolysis at 1000 °C (Figure 6a), which could be simulated by Gaussian curve fitting and assigned to SiO_4_ (−110 ppm), SiO_3_C (−74 ppm), SiO_2_C_2_ (−37 ppm), and SiC_4_ (−12 ppm) structural units [31,32], as summarized in Table 3. These results are consistent with those of previous NMR studies on PDCs [33]. However, to the best of our knowledge, no NMR studies of graphene-modified SiOC have been conducted. Intriguingly, the RGOA-1000/SiOC obtained by pyrolysis at the same temperature (i.e., 1000 °C) exhibited the characteristic signals of SiOC_3_ (7 ppm) along with SiO_4_ (−107 ppm), SiO_3_C (−75 ppm), SiO_2_C_2_ (−42 ppm), and SiC_4_ (−13 ppm) units, as shown in Figure 6b. The fraction of the SiC_4_ unit decreased, while those of the of SiO_2_C_2_ and SiO_3_C units increased (Table 3), indicating the partial conversion of the SiC_4_ unit into oxygen-containing SiO_x_C_4−x_ (SiOC_3_, SiO_2_C_2_, and SiO_3_C) units due to the interaction between RGOA and SiOC. These interactions caused Si atoms to bond with O atoms from remaining oxygen groups in reduced graphene oxide during the pyrolysis process. The ^29^Si MAS-NMR spectrum of RGOA-600/SiOC nanocomposite (Figure 6c) obtained at lower pyrolysis temperature (i.e., 600 °C) showed the characteristic signals of SiO_4_ (−110 ppm), SiO_3_C (−74 ppm), SiO_2_C_2_ (−41 ppm), and SiC_4_ (−10 ppm) units. The relative concentration of SiC_4_ unit further decreased, and that of SiO_4_ unit increased. These results further confirmed the interaction between Si atoms of SiOC and O atoms of RGOA and that graphene oxide plays a critical role in the distribution of SiO_x_C_4−x_ structural units in SiOC. The oxygen-rich SiO_x_C_4−x_ units are more prone to forming than carbon-rich SiO_x_C_4−x_ units in SiOC via the modification of graphene oxide, and the lower reduction degree of graphene oxide results in a higher concentration of oxygen-rich SiC_x_O_4−x_ units.

Figure 6d–f show the experimental and simulated ^13^C MAS NMR spectra of SiOC and RGOA-SiOC nanocomposites. In the case of SiOC sample (Figure 6d), two apparent peaks centered at 14 and 127 ppm could be assigned to sp^3^-hybridized carbon within SiO_x_C_4-x_ units and sp^2^-hybridized carbon (C_free_), respectively [34,35]. These results suggested the presence of two types of carbon in SiOC ceramic: carbon bonded to Si in SiO_x_C_4-x_ units and carbon bonded to other carbon atoms forming nanodomains of turbostratic carbon, consistent with the findings in a previous work [31]. The other three peaks located at 36, 70, and 179 ppm were spinning sidebands [31]. After the integration of SiOC into RGOA, the peak corresponded to C_free_, which was present as both turbostratic carbon and graphene sheets and was dominant in RGOA-1000/SiOC and RGOA-600/SiOC nanocomposites (Figure 6e,f), further confirming that sp^2^-hybridized reduced graphene oxide as C_free_ phase was embedded in SiOC.

Based on the above characterizations, a schematic representation of the possible nanodomain structures of the as-prepared graphene-embedded SiOC nanocomposites is shown in Figure 7. The lower thermal treatment temperature resulted in a lower reduction degree of reduced graphene oxide aerogel, which maintained a higher concentration of oxygen groups (Figure 7a,b), promoting the formation of oxygen-rich SiO_x_C_4−x_ units and improved sp^2^-hybridized C_free_ phase with graphene-SiOC nanocomposite (Figure 7c–e).

This work revealed the nanodomain structure of graphene-SiOC composites, allowing for an understanding of the structure–property relationships and providing valuable insights for the creation of high-performance graphene–PDCs nanocomposites in the areas of batteries [36,37], supercapacitors [38], catalysis [39,40], and microwave absorption and shielding [41].

## 4. Conclusions

In summary, reduced graphene oxide aerogels were prepared by pyrrole-mediated hydrothermal synthesis, followed by freeze-drying and thermal reduction. A higher thermal treatment temperature (1000 °C) resulted in a higher reduction degree of graphene oxide aerogel with fewer remaining oxygen groups. A novel reduced graphene oxide aerogel containing SiOC ceramic was prepared by the infiltration of a preceramic polymer and a subsequent high-temperature pyrolysis process. The as-prepared RGOA-SiOC presented a highly porous structure with a uniform distribution of graphene into SiOC. After the introduction of RGOA, the carbon-rich SiO_x_C_4−x_ units within SiOC were prone to transforming into oxygen-rich SiO_x_C_4-x_ units, both confirmed by Si2p XPS and ^29^Si NMR spectra, which occurred due to the interaction between the Si atoms in SiOC and the O atoms in RGOA during the pyrolysis process. The RGOA-600/SiOC nanocomposites contained a higher concentration of oxygen-rich SiO_x_C_4−x_ units due to the lower reduction degree of RGOA-600 with a higher concentration of oxygen groups. In addition to the tailorable SiO_x_C_4−x_ tetrahedral units, the free carbon phase was regulated. Long-ordered, sp^2^-hybridized graphene sheets embedded into the amorphous SiOC phase have been documented by HR-TEM, Raman, and ^13^C NMR techniques. Revealing the nanodomain structures of graphene–SiOC nanocomposites may offer a path to investigate structure-property relationships and tailor material properties for practical applications.

## Figures and Tables

**Figure 1 polymers-14-03675-f001:**
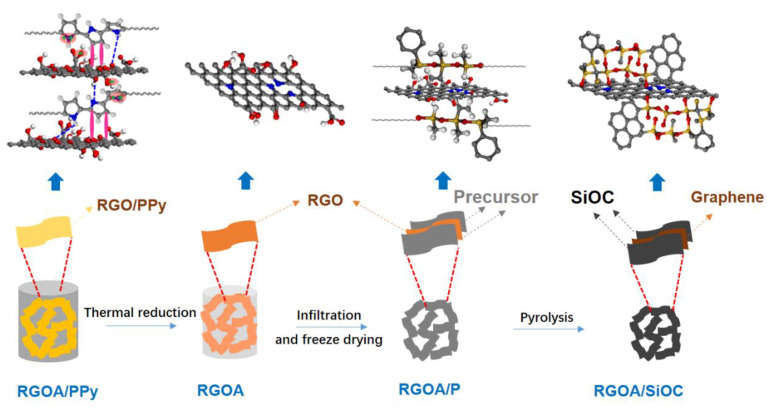
Schematic of the bottom-up fabrication process of RGOA/SiOC nanocomposites and the evolution of chemical structures and interactions. The hydrothermally synthesized RGOA/PPy aerogels were first reduced at 600–1000 °C in Ar, then infiltrated by a polymeric solution to obtain the RGOA/P precursors, which were subsequently freeze-dried and pyrolyzed at 1000 °C to obtain the RGOA/SiOC nanocomposites.

**Figure 2 polymers-14-03675-f002:**
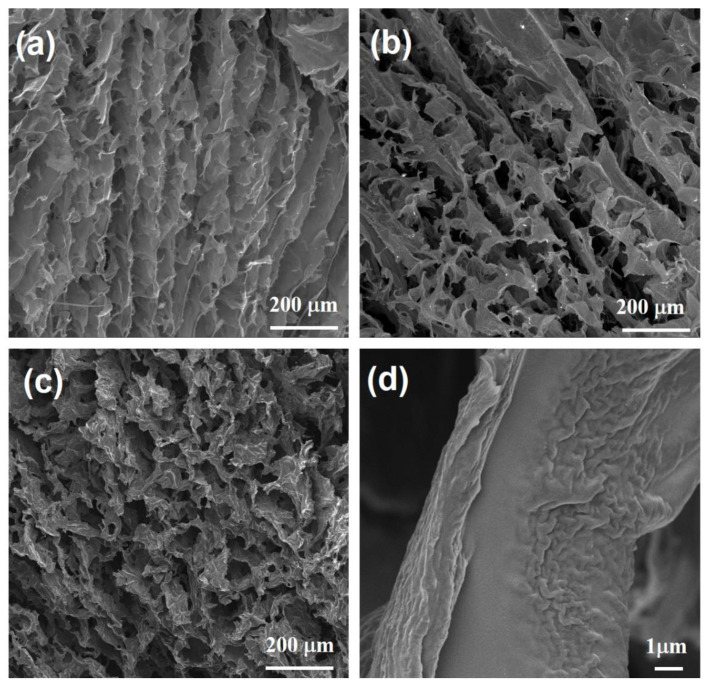
Low-magnification SEM images of (**a**) RGOA, (**b**) RGOA-1000, and (**c**) RGOA-1000/SiOC samples. (**d**) High-magnification SEM image of RGOA-1000/SiOC sample.

**Figure 3 polymers-14-03675-f003:**
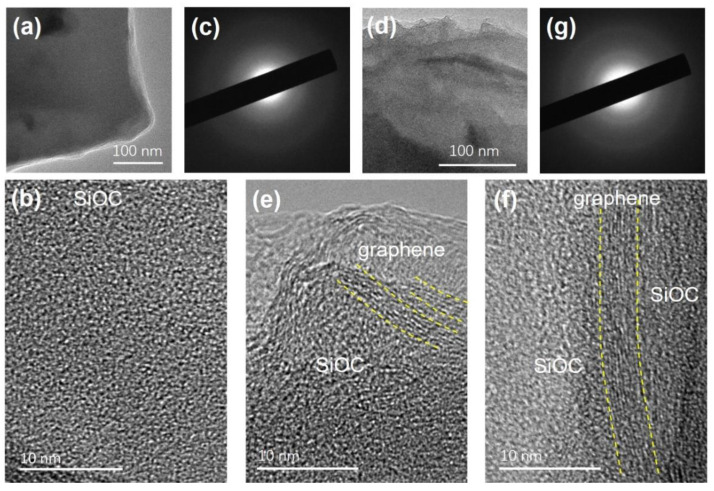
(**a**) TEM, (**b**) HRTEM, and (**c**) SAED images of SiOC sample. (**d**) TEM, (**e**,**f**) HRTEM, and (**g**) SAED images of RGOA-1000/SiOC nanocomposite.

**Figure 4 polymers-14-03675-f004:**
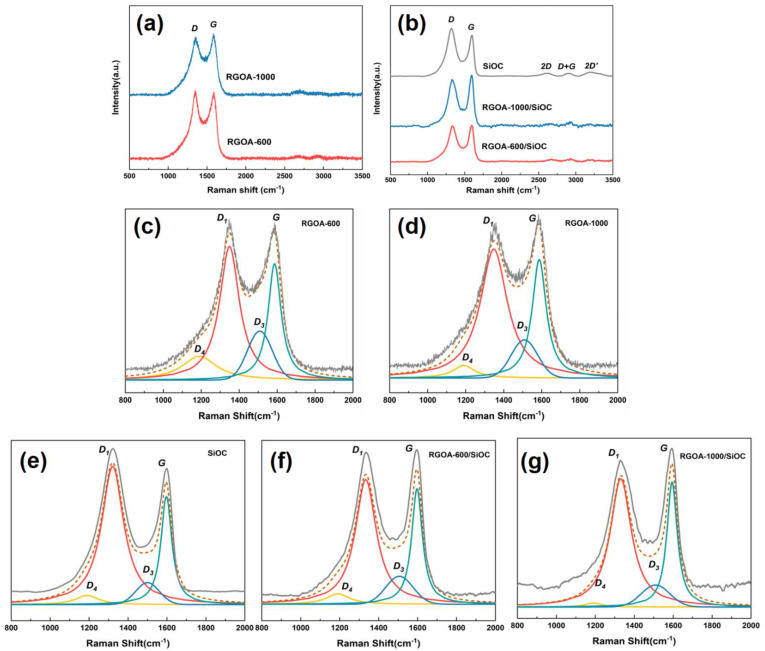
Raman spectra of (**a**) RGOA-600 and RGOA-1000 and (**b**) RGOA modified SiOC nanocomposites and SiOC. (**c**–**g**) Deconvoluted spectra of RGOA-600, RGOA-1000, SiOC, RGOA-600/SiOC, and RGOA-1000/SiOC, respectively. The experimental spectra are shown by the solid grey line on top, followed by the simulated spectra (dashed brown lines), and the individual simulation components (solid colored lines).

**Figure 5 polymers-14-03675-f005:**
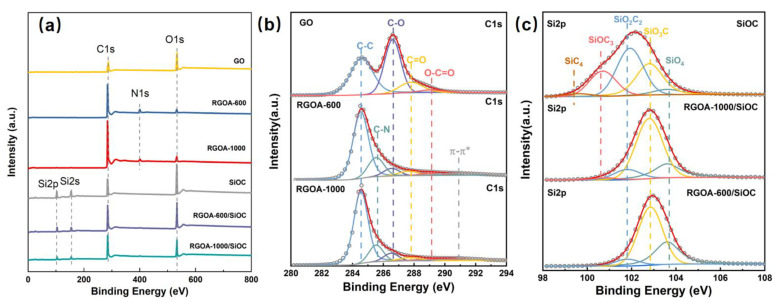
(**a**) Survey X-ray photoelectronic spectra of GO, RGOA, and RGOA-SiOC nanocomposites and SiOC. High-resolution (**b**) C1s spectra of GO, RGOA-600, and RGOA-1000 and (**c**) Si2p spectra of SiOC, RGOA-1000/SiOC, and RGOA-600/SiOC nanocomposites.

**Figure 6 polymers-14-03675-f006:**
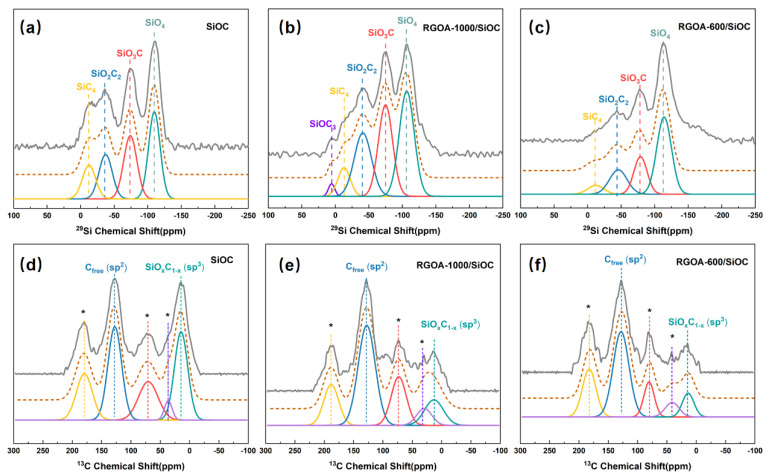
The ^29^Si and ^13^C MAS NMR spectra of (**a**,**d**) SiOC, (**b**,**e**) RGOA-1000/SiOC, and (**c**,**f**) RGOA-600/SiOC. The experimental spectra are shown by the solid grey line on top, followed by the simulated spectra (dashed brown lines), and the individual simulation components (solid colored lines).

**Figure 7 polymers-14-03675-f007:**
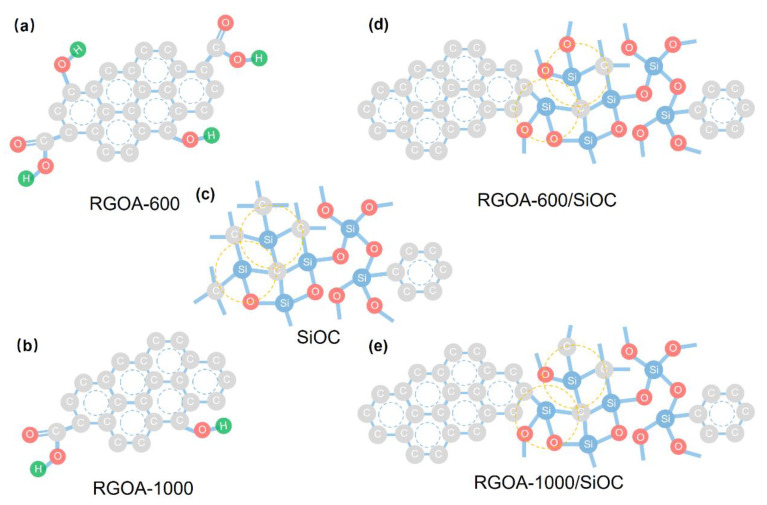
Schematic illustration of nanodomain structures of SiOC and reduced graphene oxide embedded SiOC samples. (**a**) RGOA-600, (**b**) 1000/SiOC, (**c**) SiOC, (**d**) RGOA-600/SiOC and (**e**) RGOA-1000/SiOC.

**Table 1 polymers-14-03675-t001:** Band positions, intensity ratios I(D1)/I(G) obtained from curve fitting of the Raman spectra, and lateral crystallite sizes of free carbon of the SiOC, RGOA-600 and -1000, and RGOA-600 and -1000/SiOC samples.

	D_4_, cm^−1^	D_1_, cm^−1^	D_3_, cm^−1^	G cm^−1^	I(D_1_)/I(G)	La_1_, nm	La_2_, nm
SiOC	1189	1321	1500	1597	1.27	3.90	1.48
RGOA-600	1190	1348	1509	1587	1.14	4.35	1.44
RGOA-1000	1190	1348	1509	1587	1.08	4.59	1.40
RGOA-600/SiOC	1191	1334	1507	1597	1.07	4.63	1.36
RGOA-1000/SiOC	1191	1331	1509	1594	1.03	4.81	1.33

**Table 2 polymers-14-03675-t002:** Surface elemental composition from the survey X-ray photoelectronic spectra of GO, RGOA, SiOC, and RGOA-SiOC nanocomposites.

	C (a. %)	O (at %)	N (at. %)	Si (at. %)
GO	65.6	34.4	-	-
RGOA-600	86.1	8.7	5.2	-
RGOA-1000	93.2	3.6	3.2	-
RGOA-600/SiOC	62.5	25.0	1.8	10.7
RGOA-1000/SiOC	69.0	20.6	1.8	8.6
SiOC	46.8	34.6	-	18.6

**Table 3 polymers-14-03675-t003:** Si-containing structural units in SiOC and RGOA-modified SiOC nanocomposites according to the simulation of 29Si MAS NMR spectra.

Samples		SiO_4_	SiO_3_C	SiO_2_C_2_	SiOC_3_	SiC_4_
SiOC	Chemical shift (ppm)	−110	−74	−37	-	−12
	Fraction (%) of the specific unit	34.9	29.1	20.4	-	15.6
RGOA-1000/SiOC	Chemical shift (ppm)	−107	−75	−41	7	−13
	Fraction (%) of the specific unit	35.8	29.6	24.7	1.8	8.1
RGOA-600/SiOC	Chemical shift (ppm)	−113	−78	−45	-	−11
	Fraction (%) of the specific unit	57.7	19.6	17	-	5.7

## Data Availability

Not applicable.

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
