# Peer review of "Revealing Nanodomain Structures of Bottom-Up-Fabricated Graphene-Embedded Silicon Oxycarbide Ceramics"

_polymers, 2022, doi:10.3390/polym14173675_

Round 1

Reviewer 1 Report

Manuscript No.: Polymers 2022

Date received 19 July 2022

Title: Revealing nanodomain structures of bottom-up fabricated graphene embedded silicon oxycarbide ceramics

Author: Dongxiao Hu, Gaofeng Shao, Jun Wang, Aleksander Gurlo and Maged F. Bekheet

According to the Abstract, the paper reveals the complex nanodomain structures of graphene-PDC composites, through a novel reduced graphene oxide aerogel embedded silicon oxycarbide (RGOA-SiOC) nanocomposites were bottom-up fabricated by using a 3D \reduced graphene oxide aerogel as a skeleton and followed by infiltration of ceramic precursor and high-temperature pyrolysis process.

After carefully reviewing this paper, I recommend that it:

General remarks: The paper combines interesting the structure of graphene-embedded silicon oxycarbide ceramics and the nanodomain structures of graphene-SiOC nanocomposites could offer a path to investigating the structure-property relationships and tailoring material properties for practical applications.

I consider that none of the methods can be considered original, but the motivations and goals of the experimental efforts are to be considered by the reader. Nonetheless, the paper can be of interest to the audience of Polymers, in the presented form in this work.

Author Response

Response to Reviewer 1 Comments

According to the Abstract, the paper reveals the complex nanodomain structures of graphene-PDC composites, through a novel reduced graphene oxide aerogel embedded silicon oxycarbide (RGOA-SiOC) nanocomposites were bottom-up fabricated by using a 3D \reduced graphene oxide aerogel as a skeleton and followed by infiltration of ceramic precursor and high-temperature pyrolysis process.

After carefully reviewing this paper, I recommend that it:

General remarks: The paper combines interesting the structure of graphene-embedded silicon oxycarbide ceramics and the nanodomain structures of graphene-SiOC nanocomposites could offer a path to investigating the structure-property relationships and tailoring material properties for practical applications.

I consider that none of the methods can be considered original, but the motivations and goals of the experimental efforts are to be considered by the reader. Nonetheless, the paper can be of interest to the audience of Polymers, in the presented form in this work.

Response: We would like to express our sincere gratitude to you for your efforts in reviewing our manuscript, and the recognition for this research work.

Reviewer 2 Report

In my opinion, this work is of obvious importance to the designing and creating a novel reduced graphene oxide aerogel containing SiOC ceramic. The article presents convincing evidence for the synthesis of a novel material, while sufficiently investigating its properties and analyzing its structure. In my opinion, the paper is quite detailed and provides all the comprehensive data. I do not consider it necessary to make any significant criticisms. There is no doubt that this paper is very useful for the nanomaterials communities. I think that the novelty of this paper is sufficient to be published in Polymers.

A few remarks:

Recently, hybrid polymeric structures that include not only graphene but also carbon nanotubes (in particular, silicone composites with CNT/graphene hybrid fillers) have shown great efficiency. How complementary are they to silicon oxycarbide ceramics?

Non-published materials include SEM images (Fig 2c, 2d) from 2018. Was the research really done in 2018?

The captions to the figures (1, 2, 3) are actually very brief and insufficiently informative. For example, how the microphotographs were obtained from both the text and the caption to the figure can only be guessed.

A relatively detailed report is presented, but it would have been helpful to explicitly indicate promising applications for the RGOA-SiOC.

I think it would be helpful to mention the following references in the introduction to this article:

https://pubs.acs.org/doi/10.1021/acsami.6b05032

https://pubs.acs.org/doi/10.1021/acsami.0c12376

and a recent review https://link.springer.com/article/10.1007/s40145-021-0562-2

Author Response

Response to Reviewer 2 Comments

In my opinion, this work is of obvious importance to the designing and creating a novel reduced graphene oxide aerogel containing SiOC ceramic. The article presents convincing evidence for the synthesis of a novel material, while sufficiently investigating its properties and analyzing its structure. In my opinion, the paper is quite detailed and provides all the comprehensive data. I do not consider it necessary to make any significant criticisms. There is no doubt that this paper is very useful for the nanomaterials communities. I think that the novelty of this paper is sufficient to be published in Polymers.

A few remarks:

Point 1. Recently, hybrid polymeric structures that include not only graphene but also carbon nanotubes (in particular, silicone composites with CNT/graphene hybrid fillers) have shown great efficiency. How complementary are they to silicon oxycarbide ceramics?

Response 1: Silicon oxycarbide is an amorphous ceramic that is readily formed from preceramic polymers with varying stoichiometries. At an atomic scale, SiOC is comprised of short-range ordered SiO4−xCx tetrahedra alongside a phase of free carbon. The development of SiOC is motivated by their remarkable chemical stability, versatile processing routes and tunable properties of electrical conductivity, permittivity, and thermal conductivity. The free carbon phase plays a critical role in the microstructure and its associated properties of SiOC. For example, CNT and graphene act as conductive fillers and are good for the enhancement of the overall electrical conductivity. The as-prepared hybrid polymeric structures show promising applications in batteries and microwave absorption and shielding. On the other hand, CNT and graphene also can be used as low-dimensional fillers to enhance the mechanical properties of bulk SiOC materials.

Point 2. Non-published materials include SEM images (Fig 2c, 2d) from 2018. Was the research really done in 2018?

Response 2: Yes, the SEM images were obtained in 2018 at TU Berlin.

Point 3. The captions to the figures (1, 2, 3) are actually very brief and insufficiently informative. For example, how the microphotographs were obtained from both the text and the caption to the figure can only be guessed.

Response 3: We revised the captions of Figures 1, 2 and 3 in the revised manuscript to become more informative as follows.

Figure 1. Schematic of the bottom-up fabrication process of RGOA modified SiOC nanocomposites and the evolution of chemical structures and interactions. The hydrothermally synthesized RGOA/PPy aerogels were first reduced at 600-1000 °C in Ar, and then infiltrated by a polymeric solution to obtain the RGOA/P precursors that were subsequently freeze-dried and pyrolyzed at 1000 °C to obtain the RGOA/SiOC nanocomposites. 

Figure 2. Low magnification SEM images of (a) RGOA, (b) RGOA-1000, and (c) RGOA-1000/SiOC samples. (d) High magnification SEM image of RGOA-1000/SiOC sample.

Figure 3. (a) TEM, (b) HRTEM and (c) SAED images of SiOC sample. (d) TEM, (e, f) HRTEM and (g) SAED images of RGOA-1000/SiOC nanocomposite.

Point 4. A relatively detailed report is presented, but it would have been helpful to explicitly indicate promising applications for the RGOA-SiOC.

Response 4: The promising applications such as batteries, supercapacitors, catalysis, and microwave absorption and shielding have been explicitly indicated in the last paragraph of the section of Results and discussion as follows:

This work revealed the nanodomain structure of graphene-SiOC composites, allowing understanding the structure-properties relationship, and providing valuable insights for the creation of high-performance graphene-PDCs nanocomposites in the areas of batteries [36-37], supercapacitors [38], catalysis [39], and microwave absorption and shielding [40].

Point 5. I think it would be helpful to mention the following references in the introduction to this article:

https://pubs.acs.org/doi/10.1021/acsami.6b05032

https://pubs.acs.org/doi/10.1021/acsami.0c12376

and a recent review https://link.springer.com/article/10.1007/s40145-021-0562-2 

Response 5: Thank you for the recommendation of these related and high-quality references. We have reviewed in the revised manuscript.

Reviewer 3 Report

The paper “Revealing nanodomain structures of bottom-up fabricated graphene embedded silicon oxycarbide ceramics” by Hu et al., is interesting for the science of new materials. This paper contains all the main sections and all of them are well described and commented. However there are some questions that must be solved and included in order to obtain a better knowledge of the prepared materials as well as their characteristics. These are:

1.- It is well described how is prepared the main material by vacuum infiltration of a methylphenylvinylhydrogen polysiloxane, however there is not information about the initial weights of both RGOA and polysiloxane, the final weight of the RGOA-P, the final weight of the material after pyrolysis at 600 and 1000 ºC. All of these data are needed to know.

2.- During the infiltration and pyrolysis processes are not described the porosities of the corresponding materials. It is necessary to know the initial and final porosities, densities and theoretical one. And, if it is possible, it would be interesting the specific surface areas with micro-meso and macro porosities. Because it is given the Raman and XPS spectra of the SiOC material, it would be interesting to give the porosity and density of this material as well.

3.- There is a wide study of the chemical composition of the final materials carried out by XPS, however and as the authors indicate, this study only corresponds to the surface chemical composition but not the bulk composition. Then it is necessary to give the bulk chemical composition determined by elemental analysis of C, O, Si and probably H (and N if it is possible). The bulk chemical composition permits a well knowledge of the final material.

4.- In Table 1 there are given the La values for SiOC, RGOA-600/SiOC and RGOA-1000/SiOC and such values are similar, however the main La values of the RGOA and RGOA-600 and RGOA-1000 are not given and these values are very important in order to compare the effect of SiOC on La. These values must be included.

Beside these questions, there are other minor ones such as:

a)      It would be interesting (if possible) to add the Raman spectrum of GO and RGOA

b)      In equation (1) the C(lambda) constant is not described.

c)      TK (in TK-correlation) must be described or a new reference must be included.

d)      In page 3, second paragraph (3. Results and discussion) appears “directionduring” that must be separated.

e)      In some cases appears La and others in italic format. All of them must be italic.  

Author Response

Response to Reviewer 3 Comments

The paper “Revealing nanodomain structures of bottom-up fabricated graphene embedded silicon oxycarbide ceramics” by Hu et al., is interesting for the science of new materials. This paper contains all the main sections and all of them are well described and commented. However there are some questions that must be solved and included in order to obtain a better knowledge of the prepared materials as well as their characteristics. These are:

Point 1: It is well described how is prepared the main material by vacuum infiltration of a methylphenylvinylhydrogen polysiloxane, however there is not information about the initial weights of both RGOA and polysiloxane, the final weight of the RGOA-P, the final weight of the material after pyrolysis at 600 and 1000 ºC. All of these data are needed to know.

Response 1: In this work, the concertation of polysiloxane/TBA is 25mg/mL. The final weight ratios (initial weight of RGOA/the weight of RGOA-SiOC nanocomposite) of RGOA in RGOA-600/SiOC and RGOA-1000/SiOC are 22.3 wt% and 20.5 wt%. We added this information in the experimental section of the revised manuscript.

Point 2:. During the infiltration and pyrolysis processes are not described the porosities of the corresponding materials. It is necessary to know the initial and final porosities, densities and theoretical one. And, if it is possible, it would be interesting the specific surface areas with micro-meso and macro porosities. Because it is given the Raman and XPS spectra of the SiOC material, it would be interesting to give the porosity and density of this material as well.

Response 2: We thank the reviewer for their interesting suggestions. The main aim of this work is to focus on revealing the nanodomain structure of graphene oxide modified SiOC that are prepared by using a reduced graphene oxide aerogel as a skeleton, followed by infiltration of ceramic precursor and high-temperature pyrolysis process. Although the surface area, porosity and density of these materials might be important, the authors could not perform a lot of characterizations on the materials due to the very limited amount of produced sample (i.e., 50 mg). Thus, we focused on using XPS, Raman, NMR, TEM characterizations to achieve the main goal of this work. However, we will consider all the aforementioned measurements by the reviewer in our coming works focusing on the applications of the obtained nanocomposites for battery and catalysis applications.

Point 3. There is a wide study of the chemical composition of the final materials carried out by XPS, however and as the authors indicate, this study only corresponds to the surface chemical composition but not the bulk composition. Then it is necessary to give the bulk chemical composition determined by elemental analysis of C, O, Si and probably H (and N if it is possible). The bulk chemical composition permits a well knowledge of the final material.

Response 3: We thank the reviewer for their interesting suggestions.  Unfortunately, most of the elemental analysis techniques might require a high sample weight (i.e., over 500 mg). Thus, it is very difficult to perform these measurements on our samples due to the limited weight of the obtained RGOA-SiOC samples (i.e., below 50 mg). Moreover, since our work focuses on the microstructure of RGOA-SiOC nanocomposite, especially at the nanoscale and molecular scale, XPS was sufficient to study the surface chemical compositions at that scale.

Point 4: In Table 1 there are given the La values for SiOC, RGOA-600/SiOC and RGOA-1000/SiOC and such values are similar, however the main La values of the RGOA and RGOA-600 and RGOA-1000 are not given and these values are very important in order to compare the effect of SiOC on La. These values must be included.

Response 4: The deconvoluted spectra of RGOA-600, RGOA-1000 are displayed and the La values of RGOA-600 and RGOA-1000 are included in Table 1.

Figure 4. Raman spectra of (a) RGOA-600 and RGOA-1000 and (b) RGOA modified SiOC nanocomposites and SiOC. Deconvoluted spectra of RGOA-600, RGOA-1000, SiOC, RGOA-600/SiOC and RGOA-1000/SiOC are displayed in (c)-(g). The experimental spectra are shown by the solid grey line on top, followed by the simulated spectra (dashed brown lines), and the individual simulation components (solid colored lines).

Table 1. Bands positions, intensity ratio I(D1)/I(G) obtained from curve fitting of the Raman spectra and lateral crystallite size of free carbon of the SiOC, RGOA-600 and -1000, and RGOA-600 and -1000/SiOC samples

D4, cm-1

D1, cm-1

D3, cm-1

G cm-1

I(D1)/I(G)

La1, nm

La2, nm

SiOC

1189

1321

1500

1597

1.27

3.90

1.48

RGOA-600

1190

1348

1509

1587

1.14

4.35

1.44

RGOA-1000

1190

1348

1509

1587

1.08

4.59

1.40

RGOA-600/SiOC

1191

1334

1507

1597

1.07

4.63

1.36

RGOA-1000/SiOC

1191

1331

1509

1594

1.03

4.81

1.33

Point 5. Beside these questions, there are other minor ones such as:

  1. a) It would be interesting (if possible) to add the Raman spectrum of GO and RGOA
  2. b) In equation (1) the C(lambda) constant is not described.
  3. c) TK (in TK-correlation) must be described or a new reference must be included.
  4. d) In page 3, second paragraph (3. Results and discussion) appears “directionduring” that must be separated.
  5. e) In some cases appears La and others in italic format. All of them must be italic.

Response 5: We checked and revised the above points in the revised manuscript.

Round 2

Reviewer 3 Report

This new manuscript has been improved respect to the first one and most of the questions have been answered but there remains two main questions hasn´t yet. One of them is the elemental composition determined by chemical analysis (not by XPS) and the other one is what is the yield after pyrolysis at 600 and at 1000 ºC for the prepared graphene-polymer samples? With the first question a highe knowledge of the chemistry of the prepared materials can be obtained and, from the second question it can be known the weight loss after pyrolysis when compared with the preceramic polymer itself.

Author Response

We thank the reviewer for their suggestions again. Although the yield of preceramic polymer with RGOA and the elemental composition of final RGOA-SiOC samples could increase the value of our work, unfortunately, we can not perform these measurements and characterizations at the moment because the sample materials and measurement techniques are not accessible due to the lock-down of our labs in China because of COVID-19. However, all these characterizations and measurements will be considered in our next work focusing on the applications of these materials for catalytic and energy storage applications as these materials' characteristics play very important roles in such applications.